# GTD-LLM: A Plug-and-Play LLM Reasoning Module for Gaze Target Detection

## Abstract

Gaze target detection is an important task in computer vision, aiming to predict where people in an image are looking. In our view, this task not only contains explicit image features, but also implies a large amount of prior knowledge about the correlations between human visual attention and daily activities. However, existing gaze target methods rely entirely on visual modality information to detect salient objects along the gaze direction, limiting their generalization in challenging scenarios such as activity-related, long-tailed, small-sized, or long-distance gaze targets. Inspired by the great success of LLM technology, we break away from the traditional pure-visual approaches and propose GTD-LLM, the first plug-and-play LLM reasoning module for gaze target detection in visual scenes, providing a new paradigm for traditional pure-visual approaches. Our GTD-LLM module can be plug-and-play integrated with any existing gaze target visual models and directly bring them universal performance improvements, simultaneously demonstrating strong generalizability and effectiveness. In our GTD-LLM module, we design a novel prompt engineering method GTD-Prompt, to guide LLMs like GPT-4 to perform logical reasoning on possible gaze targets, without the need for any training or fine-tuning. The proposed GTD-Prompt method can also be easily extended to downstream tasks by simply adjusting the corresponding task prompt words, further illustrating its versatility.

## 1 Introduction

Gaze target detection is an important task in computer vision, aiming to predict where people in an image are looking Recasens et al. (2015). Besides, it also extends to multiple downstream tasks, *e.g.*, shared attention detection Fan et al. (2018) which predicts the shared gaze target of multiple people, and mutual gaze detection Marin-Jimenez et al. (2019) which distinguishes whether two people are looking at each other. These tasks have significant value in understanding human visual attention.

In our view, the gaze target detection task not only contains explicit image features, but also implies a large amount of prior knowledge about the correlations between human visual attention and daily activities. However, existing gaze target detection methods Chong et al. (2020); Fang et al. (2021); Bao et al. (2022) rely entirely on visual modality information to detect salient objects along the gaze direction, limiting their generalization in challenging scenarios, *e.g.*, activity-related, long-tailed, small-sized, or long-distance gaze targets. Recently, large language models (LLMs) achieve great success in natural language processing (NLP) and are also increasingly introduced into computer vision tasks, *e.g.*, image captioning Li et al. (2023), object detection Wang et al. (2024b), visual question answering Liu et al. (2024), *etc.*. Compared to visual models, LLMs, due to its powerful pre-training of natural language, contain a large amount of prior knowledge about human activities. Inspired by this, we break away from the traditional pure-visual approaches. We consider how to leverage the powerful logical reasoning ability of LLMs to address gaze target detection in visual scenes, and consider how to develop a plug-and-play LLM reasoning module. This module should be able to integrate with any existing gaze target visual models in a plug-and-play manner, and directly bring them universal performance improvements.

To achieve this goal, we first analyze the human thought processes in gaze target detection, and then consider how to use visual models and LLMs to simulate them. As shown in Fig. 1, the human thought processes can be broken down into three steps: object information extraction, object position analysis, and gaze target reasoning. Due to the low information density of the image itself, human

| Object Information Extraction | Object Position Analysis | Gaze Target Reasoning |

Figure 1: The human thought processes in gaze target detection.

observers will first extract the key object-level information from it, including object categories/locations and human gaze direction/body pose, *etc.*. This process is obviously suitable for simulation using visual detection models which are good at capturing detail features of images. Next, human observers will analyze the object position relationships based on the extracted information, *e.g.*, *"For a person in the image, which objects are located within his field of view (FOV)? Which are outside?"*. This step can be achieved through manually formulated rules. Finally, based on above analyses, human observers will reason *"What activities the person may be doing? Which object he may be looking at?"*. This logical reasoning process is obviously more suitable for LLMs.

Based on above analyses, we propose GTD-LLM, the first plug-and-play LLM reasoning module for gaze target detection in visual scenes, providing a new paradigm for traditional pure-visual approaches. Our GTD-LLM module directly reads the key object-level information extracted from input images through pre-trained object-level detectors, without reading raw images with low information density. This will significantly reduce the computational burden of LLMs. Our GTD-LLM module uses a specially designed prompt engineering method GTD-Prompt, to guide LLMs like GPT-4 to perform logical reasoning on possible gaze targets, without the need for any training or fine-tuning. Besides, our GTD-LLM module introduces a simple modal transformation mechanism to transform its natural language predictions into the same visual modality as the output of existing gaze target visual models. It is precisely because of our unique design of the LLM reasoning method and the input/output interfaces, that our GTD-LLM module can be plug-and-play integrated with any existing gaze target visual models, demonstrating strong generalizability. Please note that, the integrated gaze target framework is also called GTD-LLM in this paper. Our GTD-LLM framework utilizes both the logical reasoning ability of LLMs and the ability of existing gaze target visual models to capture detail features of input images. Therefore, it can bring universal and significant performance improvements to any existing gaze target visual models, especially in those challenging scenarios, *e.g.*, activity-related, long-tailed, small-sized, or long-distance gaze targets, demonstrating strong effectiveness.

In order to guide LLMs to fully mining the prior knowledge about correlations between human visual attention and daily activities, we decompose gaze target detection into a sequence of atomic-level tasks in our GTD-Prompt method based on common sense. Specifically, we design the following task flow prompts, *"What kind of scene is this image?"*, *"For each person, what are they doing?"*, *"Where are they looking?"*. These atomic-level tasks conform to human logic and are easier for LLMs to understand. Besides, we also design a series of position relationship rules to transform the extracted object-level information into structured natural language descriptions. Then, we use our task flow prompts (*i.e.*, the instruction) to guide LLMs like GPT-4 to reason the possible gaze targets from these structured object position relationships (*i.e.*, the input content) for each person in the image step by step. Our GTD-Prompt method can also be easily extended to downstream tasks, *e.g.*, shared attention detection and mutual gaze detection, by simply adjusting the corresponding task prompt words. For example, by adding *"Is there multiple people looking at the same target?"* after the original task flow prompts, we can guide LLMs to continue reasoning the shared gaze target based on the analysis results of gaze target detection. This further illustrates the versatility of our method in understanding the human visual attention in daily activities.

In summary, our main contributions are as follows:

- We propose GTD-LLM, the first plug-and-play LLM reasoning module for gaze target detection in visual scenes, providing a new paradigm for traditional pure-visual approaches.

- Our GTD-LLM module can be plug-and-play integrated with any existing gaze target visual models and bring them universal performance improvements, simultaneously demonstrating strong generalizability and effectiveness.

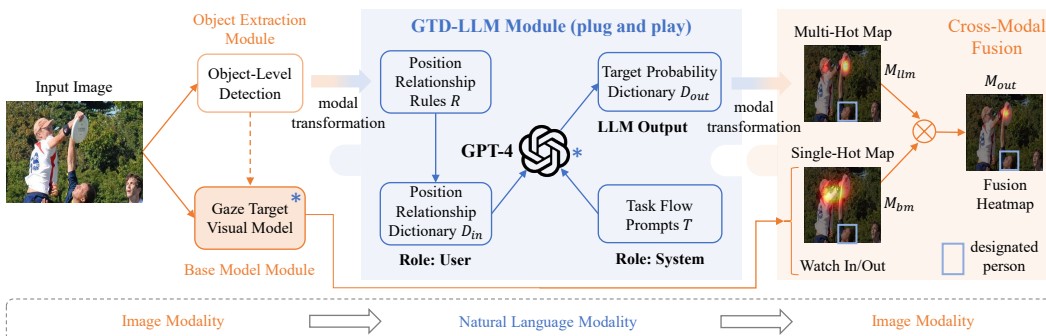

Figure 2: Overview of the plug-and-play GTD-LLM module and integrated GTD-LLM framework.

- In our GTD-LLM module, we design a novel prompt engineering method GTD-Prompt, to guide LLMs like GPT-4 to perform logical reasoning on possible gaze targets, without the need for any training or fine-tuning.
- The proposed GTD-Prompt method can also be easily extended to downstream tasks by simply adjusting the corresponding task prompt words, further illustrating its versatility.

## 2 RELATED WORK

**Gaze Target Detection.** Recasens *et al* Recasens et al. (2015) pioneered the field by introducing the GazeFollow dataset, comprising a substantial collection of images annotated with head positions and corresponding gaze targets. Chong *et al* Chong et al. (2020) extended the task to include out-of-frame scenarios, introducing a video dataset for this purpose. Tu *et al* Tu et al. (2022) extended the task to simultaneously detect all human faces and their gaze targets in a single image. Fan *et al* Fan et al. (2018) proposed the shared attention detection task, which aims to predict the shared gaze target of multiple people. Marin *et al* Marin-Jimenez et al. (2019) introduced the mutual gaze detection task, aiming to distinguish whether two people are looking at each other.

**Large Language Models.** Large Language Models have transformed NLP by demonstrating powerful abilities in language understanding and generation. GPT-3 Brown et al. (2020) introduced a large-scale autoregressive model that excels at few-shot learning across various tasks. GPT-4 OpenAI (2023) extended these capabilities with a larger model architecture and improved handling of complex reasoning tasks, showcasing remarkable performance in understanding nuanced prompts and integrating multimodal data, including text and images. Compared to previous LLMs Devlin et al. (2019); Raffel et al. (2020), GPT-4 exhibits superior generalization and problem-solving abilities, especially in scenarios requiring reasoning and domain adaptation.

**Prompt Engineering in Computer Vision.** Prompt engineering has gained traction in leveraging LLMs for vision tasks. CoOp Zhou et al. (2022) extended prompt engineering by learning task-specific prompts for visual tasks, improving performance on unseen categories. Flamingo Alayrac et al. (2022) demonstrated how visual and language models can be effectively combined for tasks like image captioning and visual question answering through flexible multimodal prompts. BLIP-2 Li et al. (2023) proposed a bootstrapping technique that bridges vision-language models with large language models. These works highlight the increasing importance of prompt-based methods for unifying vision and language tasks.

## 3 METHOD

In this section, we provide a detailed introduction to the plug-and-play GTD-LLM module and the integrated GTD-LLM framework. As shown in Fig. 2, the integrated GTD-LLM framework consists of four modules: object extraction module, GTD-LLM module, base model module, and cross-modal fusion mechanism. Fig. 3 shows an example of the proposed GTD-Prompt method. We chose GPT-4 as the LLM for logical reasoning in our experiments.

### 3.1 OBJECT EXTRACTION MODULE

We use the pre-trained MM-GroundingDINO Zhao et al. (2024) to detect objects of LVIS categories Gupta et al. (2019) from the input image. We also use the pre-trained OpenPose Cao et al. (2017)

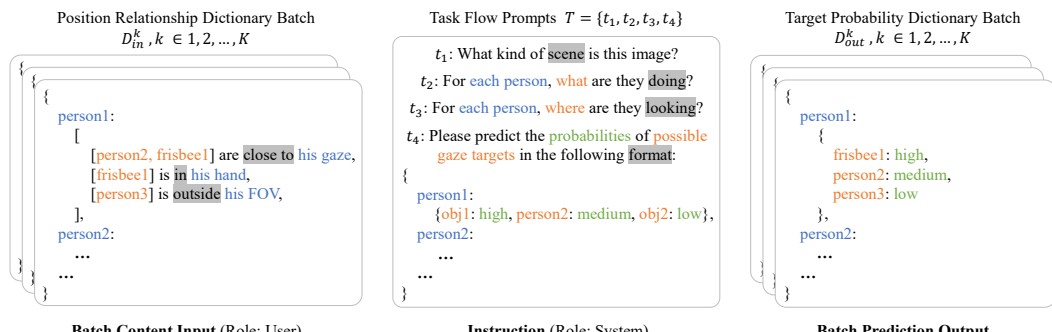

Figure 3: Example of the proposed prompt engineering method GTD-Prompt.

to detect human body pose (including face, hands, feet), and L2cs-net Abdelrahman et al. (2023) to estimate gaze direction. Among them, we define all detected human instances as $\{o_1, o_2, ..., o_M\}$, all detected object instances as $\{o_{M+1}, o_{M+2}, ..., o_N\}$. In this way, all detected human and object instances are represented as $\{o_i \mid i = 1, 2, ..., N\}$. Then, we calculate the angle and distance values between each human and other objects/humans based on their coordinates. For each human $\{o_i \mid i = 1, 2, ..., M\}$, we calculate the angle value $a_{i,j}$ between his gaze vector and the direction vector from his face center to the center/face of each other object/human $\{o_j \mid j \in \{1, 2, ..., N\} \cap j \neq i\}$. We also calculate the minimum distance value $d_{i,j}$ between his hands and all points in each other object/human.

### 3.2 GTD-LLM MODULE (PLUG AND PLAY)

First, we transform the calculated angle and distance values into human-centered position relationships described by natural language, through a set of position relationship rules. Then, we structure these natural language descriptions into position relationship dictionaries, and feed these dictionaries as content into GPT-4 in a batch format. Next, we use the specially designed task-flow prompt sequence, which follows the human thought processes, as the instruction to guide GPT-4 to reason the possible gaze targets from the batch content input. Finally, we let GPT-4 structure its predictions into target probability dictionaries, for the convenience of subsequent batch processing and integration with the output of existing gaze target visual models. The reason why we do not let GPT reason the out-of-frame classification task is provided in the appendix.

**Position Relationship Rules.** The reason why we do not directly input the detected object coordinates or the calculated angle/distance values into GPT-4 is provided in the appendix. We pre-define a set of position relationship rules $R = \{r_a, r_d\}$ to transform these angle/distance values into the human-centered position relationship descriptions which conforms to human expression habits. These natural language descriptions are easier for GPT-4 to understand. For each human $\{o_i \mid i = 1, 2, ..., M\}$, the angular relationship descriptions between him and other objects/humans $\{o_j \mid j \in \{1, 2, ..., N\} \cap j \neq i\}$ are created by the angular relationship rule $r_a$ as follows,

$$r_a(o_i, o_j, a_{i,j}) = \begin{cases} \text{``for } o_i, \ o_j \text{ is close to his gaze''}, & |a_{i,j}| \leq \alpha_1 \\ \text{``for } o_i, \ o_j \text{ is within his FOV''}, & \alpha_1 < |a_{i,j}| \leq \alpha_2 \ , \\ \text{``for } o_i, \ o_j \text{ is outside his FOV''}, & |a_{i,j}| > \alpha_2 \end{cases} \quad (1)$$

where $\alpha_1$ and $\alpha_2$ are thresholds to distinguish whether other objects/humans are located close to his gaze, within his FOV, or outside his FOV. Through experiments, we set $\alpha_1$ to 15° and $\alpha_2$ to 45°. The distance relationship descriptions are created by the distance relationship rule $r_d$,

$$r_d(o_i, o_j, d_{i,j}) = \begin{cases} \text{``for } o_i, \ o_j \text{ is in his hand''}, & d_{i,j} = 0 \\ \text{``for } o_i, \ o_j \text{ is near his hand''}, & d_{i,j} \leq \beta \ , \\ \text{``for } o_i, \ o_j \text{ is far from his hand''}, & d_{i,j} > \beta \end{cases} \quad (2)$$

where $\beta$ is the threshold to distinguish whether other objects are located in the human's hand, near his hand, or far from his hand. We set $\beta$ to 0.5 times the width of the human's face. We can also use the similar method to generate the position relationship descriptions between human feet and other objects. Compared to quantitative angle/distance values, these natural language descriptions are easier for GPT-4, which has powerful natural language pre-training, to understand.

**Position Relationship Dictionary.** The reason why we need to structure the above position relationship descriptions is provided in the appendix. By using these position relationship rules,

for each human $\{o_i \mid i = 1, 2, ..., M\}$, we generate a set of position relationship descriptions $\{\{r_a(o_i, o_j, a_{i,j}), r_d(o_i, o_j, d_{i,j})\} \mid j \in \{1, 2, ..., N\} \cap j \neq i\}$ between him and other objects/humans. Among them, we merge the similar position relationships into one description. As shown in Fig. 3, for *"person1"*, the objects *"person2"* and *"frisbee1"* are both located close to his gaze. Therefore, we merge them into one description *"For person1, [person2, frisbee1] are close to his gaze."*. Through this mechanism, each human will only have a maximum of six position relationship descriptions, including a maximum of three merged angular relationship descriptions (defined as the set $D_a$), and a maximum of three merged distance relationship descriptions (defined as the set $D_d$), no matter how many objects and humans there are in the image.

As shown in Fig. 3, for an image sample, we take each human $\{o_i \mid i = 1, 2, ..., M\}$ as the keys and his position relationship descriptions $\{D_a^i, D_d^i\}$ as the corresponding values, creating a structured position relationship dictionary $D_{in} = \{h_i : \{D_a^i, D_d^i\} \mid i = 1, 2, ..., M\}$. Then, we obtain a dictionary batch $D_{in}^k, k = 1, 2, ..., K$, corresponding to the image batch $I_k, k = 1, 2, ..., K$. Through the API of GPT-4, we feed this dictionary batch as the content input ('Role: User'). Through the above operations, we can effectively control the length of the input content of GPT-4, and make batch processing of these data more convenient.

**Task Flow Prompts.** The reason why we need to decompose gaze target detection into atomic-level tasks is provided in the appendix. As shown in Fig. 3, we design a coarse-to-fine task-flow prompt sequence $T = \{t_1, t_2, t_3, t_4\}$, which follows the human thought processes. We use these task flow prompts as the instruction ('Role: System'), to guide GPT-4 to reason the possible gaze targets from the input position relationship dictionary batch. Specifically, through the instruction $t_1$, we first guide GPT-4 to analyze what scene each image represents. Then, we use the instruction $t_2$ to guide GPT-4 to analyze what activities each person in the image may be doing. Next, through the instruction $t_3$, we guide GPT-4 to reason which objects they may be looking at based on previous analyses. Finally, we use the instruction $t_4$ to let GPT-4 structure its prediction of gaze targets.

**Target Probability Dictionary.** Under the guidance of $\{t_1, t_2, t_3\}$, GPT-4 will output its analysis processes and prediction results in the form of natural language descriptions, *e.g.*, *"This is a scene of a group of people playing frisbee. For person1, the frisbee1 is located in his hand and close to his gaze, so he is catching it. According to common sense, when a person is catching a frisbee, he is highly likely watching it. Therefore, the most likely gaze target of person1 is the frisbee1."* Although these analyses conform to human logic, the natural language descriptions are difficult to batch process. Therefore, we need to guide GPT-4 to structure its natural language predictions. As shown in Fig. 3, we use the instruction $t_4$ to let GPT-4 make a multi-hot prediction of gaze targets for each human $\{o_i \mid i = 1, 2, ..., M\}$ in the image, *i.e.*, make GPT-4 predict the probabilities $p_{i,j}$ of each other object/human $\{o_j \mid j \in \{1, 2, ..., N\} \cap j \neq i\}$ becoming his real gaze target. Due to the difficulty of quantitatively predicting these probabilities for GPT-4, we instruct GPT-4 to qualitatively predict them in the following manner, $p \in \{$*"high"*, *"medium"*, *"low"*$\}$. Then, for each human $\{o_i \mid i = 1, 2, ..., M\}$, we take other objects/humans $\{o_j \mid j \in \{1, 2, ..., N\} \cap j \neq i\}$ as the keys and their probabilities $p_{i,j}$ as the corresponding values, creating a person-level target probability dictionary $P_{out}^i$. For an image sample, we take each human $\{o_i \mid i = 1, 2, ..., M\}$ in the image as the keys and their person-level target probability dictionaries $P_{out}^i$ as the corresponding values, creating an image-level target probability dictionary $D_{out} = \{o_i : P_{out}^i \mid i = 1, 2, ..., M\}$. Finally, we obtain a dictionary batch $D_{out}^k, k = 1, 2, ..., K$, corresponding to the input image batch $I_k, k = 1, 2, ..., K$.

## 3.3 BASE MODEL MODULE

Any existing gaze target visual models can be used as our base model module. They directly take the original image as input. For some models Fang et al. (2021); Yang et al. (2024), it is also necessary to use the extracted object-level information, *e.g.*, face location, gaze direction, *etc.*, as input. The base model module will create a corresponding gaze target heatmap $M_{bm}^i$ for each human $\{o_i \mid i = 1, 2, ..., M\}$ in the image.

## 3.4 CROSS-MODAL FUSION MECHANISM

**Modal Transformation.** Although the natural language predictions of GPT-4 is structured into dictionaries, it is still difficult to directly integrate them with the target heatmap generated by the base model module. Therefore, we design a novel modal transformation mechanism to create a multi-hot target heatmap for each human in the image from their corresponding target probability dictionaries. Specifically, for each human $\{o_i \mid i = 1, 2, ..., M\}$, we set a two-dimensional Gaussian

Table 1: Evaluation in all COCO-category gaze targets in the GazeFollow test set. 'Sports Ball', ..., 'Kite': activity-related categories. 'COCO-LT': long-tailed categories. 'COCO-All': all categories. '+ GTD-LLM': integrating existing gaze target visual models with our GTD-LLM module.

| Methods | Sports Ball | | Cell Phone | | Frisbee | | Book | | Kite | | COCO-LT | | COCO-All | |
|---|---|---|---|---|---|---|---|---|---|---|---|---|---|---|
| | RR ↑ | Dist. ↓ | RR ↑ | Dist. ↓ | RR ↑ | Dist. ↓ | RR ↑ | Dist. ↓ | RR ↑ | Dist. ↓ | RR ↑ | Dist. ↓ | RR ↑ | Dist. ↓ |
| Video Chong et al. (2020) | 0.622 | 0.114 | 0.604 | 0.093 | 0.451 | 0.141 | 0.750 | 0.098 | 0.621 | 0.178 | 0.529 | 0.155 | 0.798 | 0.134 |
| **Video + GTD-LLM** | **0.711** | **0.084** | **0.698** | **0.068** | **0.697** | **0.093** | **0.844** | **0.084** | **0.793** | **0.126** | **0.593** | **0.138** | **0.821** | **0.124** |
| Improvement Ratio | 14% | 26% | 16% | 27% | 55% | 34% | 13% | 15% | 28% | 29% | 12% | 11% | 3% | 8% |
| Fang Fang et al. (2021) | 0.581 | 0.097 | 0.606 | 0.088 | 0.492 | 0.127 | 0.752 | 0.089 | 0.561 | 0.180 | 0.562 | 0.148 | 0.815 | 0.120 |
| **Fang + GTD-LLM** | **0.742** | **0.074** | **0.728** | **0.060** | **0.730** | **0.086** | **0.861** | **0.074** | **0.779** | **0.131** | **0.625** | **0.120** | **0.838** | **0.113** |
| Improvement Ratio | 28% | 24% | 20% | 32% | 48% | 32% | 15% | 17% | 39% | 27% | 11% | 19% | 3% | 6% |
| HGTTR Tu et al. (2022) | 0.523 | 0.065 | 0.408 | 0.027 | 0.411 | 0.073 | 0.581 | 0.056 | 0.241 | 0.073 | 0.271 | 0.102 | 0.461 | 0.099 |
| **HGTTR + GTD-LLM** | **0.579** | **0.064** | **0.500** | **0.025** | **0.500** | **0.069** | **0.628** | **0.054** | **0.299** | **0.067** | **0.346** | **0.100** | **0.473** | **0.098** |
| Improvement Ratio | 11% | 2% | 23% | 7% | 22% | 6% | 8% | 4% | 24% | 8% | 28% | 2% | 3% | 1% |
| Tonini Tonini et al. (2023) | 0.440 | 0.057 | 0.496 | 0.039 | 0.427 | 0.062 | 0.567 | 0.038 | 0.422 | 0.057 | 0.489 | 0.072 | 0.582 | 0.068 |
| **Tonini + GTD-LLM** | **0.620** | **0.055** | **0.659** | **0.038** | **0.606** | **0.059** | **0.967** | **0.035** | **0.618** | **0.048** | **0.593** | **0.068** | **0.612** | **0.065** |
| Improvement Ratio | 41% | 4% | 33% | 3% | 42% | 5% | 71% | 8% | 46% | 16% | 21% | 6% | 5% | 4% |
| Yang Yang et al. (2024) | 0.740 | 0.074 | 0.725 | 0.061 | 0.728 | 0.086 | 0.859 | 0.075 | 0.772 | 0.133 | 0.622 | 0.122 | 0.835 | 0.115 |
| Yang* | 0.578 | 0.098 | 0.604 | 0.089 | 0.486 | 0.129 | 0.750 | 0.090 | 0.552 | 0.183 | 0.556 | 0.150 | 0.812 | 0.122 |
| **Yang* + GTD-LLM** | **0.746** | **0.072** | **0.729** | **0.060** | **0.736** | **0.084** | **0.863** | **0.073** | **0.784** | **0.129** | **0.628** | **0.119** | **0.841** | **0.112** |
| Improvement Ratio | 29% | 27% | 21% | 33% | 51% | 35% | 15% | 19% | 42% | 30% | 13% | 21% | 4% | 8% |

distribution for each other object/human $\{o_j \mid j \in \{1, 2, ..., N\} \cap j \neq i\}$ in the image space,

$$Gauss_{i,j} = f(\boldsymbol{x}; \boldsymbol{\mu}_j, \sigma_j^2, A_{i,j}) = A_{i,j} \cdot \frac{1}{2\pi\sigma_j^2} \exp\left(-\frac{(\boldsymbol{x} - \boldsymbol{\mu}_j)^T (\boldsymbol{x} - \boldsymbol{\mu}_j)}{2\sigma_j^2}\right), \sigma_j^2 = \left(\frac{r_j}{2}\right)^2 \quad (3)$$

where $\boldsymbol{x} = (x, y)$ denotes any point in the image, $r_j$ is the radius of the other object/human $o_j$, $\boldsymbol{\mu} = (x_c^j, y_c^j)$ represents the center point of $o_j$, $A_{i,j}$ is the peak value of the Gaussian distribution. Through experiments, we set $A_{i,j} \in \{1.0, 0.3, 0.1\}$ corresponding to the predicted target probabilities $p_{i,j} \in \{\text{"high"}, \text{"medium"}, \text{"low"}\}$, respectively. For each human $\{o_i \mid i = 1, 2, ..., M\}$, we add up all the other objects'/humans' Gaussian distributions $\{Gauss_{i,j} \mid j \in \{1, 2, ..., N\} \cap j \neq i\}$ in the image space to obtain a multi-hot target heatmap $M_{llm}^i$, and set all values greater than 1 in it to 1,

$$M_{llm}^i = min(\Sigma_{j=1}^N Gauss_{i,j}, 1). \quad (4)$$

**Heatmap Fusion.** For each human $\{o_i \mid i = 1, 2, ..., M\}$, we directly perform pixel multiplication on the multi-hot target heatmap $M_{llm}^i$ output by our GTD-LLM module, and the single-hot heatmap $M_{bm}^i$ output by the base model module, to obtain the final fusion heatmap $M_{out}^i$,

$$M_{out}^i = norm(M_{llm}^i + b_{llm}) \cdot M_{bm}^i, \quad (5)$$

where $b_{llm}$ denotes the bias added to the heatmap $M_{llm}^i$ to enhance its fault tolerance. $norm()$ represents the normalization operation. We use the maximum value point in the fusion heatmap $M_{out}^i$ as the predicted gaze point of human $o_i$, and the object/human corresponding to that point as the predicted object-level gaze target. Through this fusion mechanism, our GTD-LLM module can be plug-and-play integrated with any existing gaze target visual models.

## 4 EXPERIMENT

### 4.1 EXPERIMENT SETTING

**Implementation Details.** In order to verify that our GTD-LLM module can bring universal performance improvements to any existing gaze target visual methods, we use all recent open-source models Chong et al. (2020); Fang et al. (2021); Tu et al. (2022); Tonini et al. (2023) as our base model module separately without any additional training. Among them, the experimental results of HGTTR Tu et al. (2022) are obtained from the unofficial open-source code [1]. Besides, we also reproduce the SOTA method 'Yang' Yang et al. (2024), which combines gaze target detection with HOI detection, and its variant 'Yang*', which abandons the HOI module. In our experiments, we set the batch size to 20, which means feeding 20 position relationship dictionaries corresponding to 20 input images as the input content into GPT-4 at once.

---

[1]https://github.com/francescotonini/human-gaze-target-detection-transformer

Table 2: Evaluation in long-distance (left) or small-sized (middle) gaze targets of COCO categories in the GazeFollow test set, and in complete gaze target datasets (right). $d$: $L_2$ distance between the designated human and his gaze target. $w_h$: width of the human's face. $w$: normalized width of the gaze target. The image width is considered as 1. $D_1$: GazeFollow dataset. $D_2$: VideoAttnTarget dataset. $D_1 \rightarrow D_2$: domain adaptation from the source domain $D_1$ to the target domain $D_2$. MD: minimum $L_2$ distance metric. AD: average $L_2$ distance metric.

| Methods | $d > 5w_h$ | | $d > 2w_h$ | | $w < 0.05$ | | $w < 0.2$ | | $D_1$ | | $D_2$ | $D_1 \rightarrow D_2$ |
|---|---|---|---|---|---|---|---|---|---|---|---|---|
| | RR ↑ | Dist. ↓ | RR ↑ | Dist. ↓ | RR ↑ | Dist. ↓ | RR ↑ | Dist. ↓ | MD ↓ | AD ↓ | Dist. ↓ | Dist. ↓ |
| Video Chong et al. (2020) | 0.406 | 0.242 | 0.649 | 0.152 | 0.354 | 0.135 | 0.501 | 0.131 | 0.077 | 0.137 | 0.134 | 0.146 |
| **Video + GTD-LLM** | **0.509** | **0.211** | **0.696** | **0.138** | **0.478** | **0.090** | **0.615** | **0.107** | **0.069** | **0.128** | **0.129** | **0.135** |
| Improvement Ratio | 25% | 13% | 7% | 9% | 35% | 33% | 23% | 18% | 10% | 7% | 4% | 8% |
| Fang Fang et al. (2021) | 0.464 | 0.218 | 0.665 | 0.140 | 0.336 | 0.098 | 0.498 | 0.117 | 0.067 | 0.124 | 0.108 | 0.117 |
| **Fang + GTD-LLM** | **0.577** | **0.195** | **0.710** | **0.133** | **0.465** | **0.078** | **0.614** | **0.102** | **0.060** | **0.116** | **0.105** | **0.111** |
| Improvement Ratio | 24% | 11% | 7% | 5% | 38% | 20% | 23% | 13% | 10% | 7% | 3% | 5% |
| HGTTR Tu et al. (2022) | 0.371 | 0.169 | 0.440 | 0.112 | 0.332 | 0.060 | 0.333 | 0.088 | 0.055 | 0.104 | 0.229 | 0.246 |
| **HGTTR + GTD-LLM** | **0.401** | **0.166** | **0.457** | **0.110** | **0.428** | **0.058** | **0.396** | **0.086** | **0.053** | **0.102** | **0.203** | **0.213** |
| Improvement Ratio | 8% | 2% | 4% | 2% | 29% | 3% | 19% | 2% | 4% | 2% | 11% | 13% |
| Tonini Tonini et al. (2023) | 0.296 | 0.087 | 0.501 | 0.070 | 0.294 | 0.056 | 0.381 | 0.061 | 0.029 | 0.069 | 0.102 | 0.108 |
| **Tonini + GTD-LLM** | **0.415** | **0.084** | **0.550** | **0.068** | **0.456** | **0.054** | **0.523** | **0.059** | **0.027** | **0.067** | **0.100** | **0.104** |
| Improvement Ratio | 40% | 3% | 10% | 3% | 55% | 4% | 37% | 3% | 7% | 3% | 2% | 4% |
| Yang Yang et al. (2024) | 0.566 | 0.197 | 0.704 | 0.135 | 0.456 | 0.080 | 0.609 | 0.104 | 0.061 | 0.118 | / | / |
| Yang* | 0.443 | 0.223 | 0.659 | 0.142 | 0.323 | 0.101 | 0.492 | 0.119 | 0.068 | 0.126 | 0.106 | 0.115 |
| **Yang* + GTD-LLM** | **0.585** | **0.193** | **0.713** | **0.132** | **0.474** | **0.076** | **0.617** | **0.101** | **0.059** | **0.115** | **0.102** | **0.107** |
| Improvement Ratio | 32% | 14% | 8% | 7% | 47% | 25% | 25% | 15% | 13% | 9% | 4% | 7% |

**Dataset Pre-processing.** Our GTD-LLM module predicts gaze targets at the object level. However, existing gaze target detection datasets, *e.g.*, GazeFollow Recasens et al. (2015) and VideoAttnTarget Chong et al. (2020), only label the ground-truth gaze point coordinates at the pixel level. Therefore, we pre-process the GazeFollow dataset, which contains rich scenes and gaze targets, in our experiments. For the test set, we use the pre-trained YOLOv10 Wang et al. (2024a) to detect COCO-category objects. Considering that each image sample in it contains up to 10 gaze point annotations corresponding to the designated human, we take the objects which contain at least 2 gaze points as the object-level ground truths.

**Evaluation Metrics.** We use both the object-level metric, Recall Rate (RR), and the pixel-level metric, $L_2$ Distance, to comprehensively evaluate the performance of the integrated GTD-LLM framework in the GazeFollow test set. Since our experiments aim to verify the performance improvement brought by our GTD-LLM module to existing gaze target visual models, we use the commonly used recall rate metric to represent the proportion of various difficult samples correctly predicted by the model, instead of the precision rate. Specifically, we consider the sample with the predicted gaze point located within the ground-truth gaze target as the positive case, otherwise as the negative case. The $L_2$ Distance metric denotes the $L_2$ distance between the predicted gaze point and the corresponding ground truth. Please refer to the appendix for why we abandon other metrics in our experiments.

## 4.2 EVALUATION IN COCO-CATEGORY GAZE TARGETS

As shown in Table 1, by integrating with our GTD-LLM module, all these gaze target visual models achieve universal performance improvements on all COCO-category gaze targets in the Gaze-Follow test set. Specifically, the recall rate improves by 3%–5%, and $L_2$ distance error reduces by 1%–8%. Especially in various challenging scenarios for visual models, *e.g.*, activity-related, long-tailed, small-sized, or long-distance gaze targets, the improvements are particularly significant. These results demonstrate the strong generalizability and effectiveness of our method.

**Activity-Related Category.** According to common sense, some specific categories of objects, *e.g.*, sports ball, cell phone, frisbee, book, and kite, *etc.*, often become the gaze targets of human in daily activities. By integrating with our GTD-LLM module, existing gaze target visual models achieve significant performance improvements in gaze targets of these activity-related categories. The recall rate improves by 8%–71%, and $L_2$ distance error reduces by 2%–35%. These demonstrate that our GTD-LLM module can effectively overcome the shortcomings of existing visual models in lacking the prior knowledge of correlations between human visual attention and daily activities.

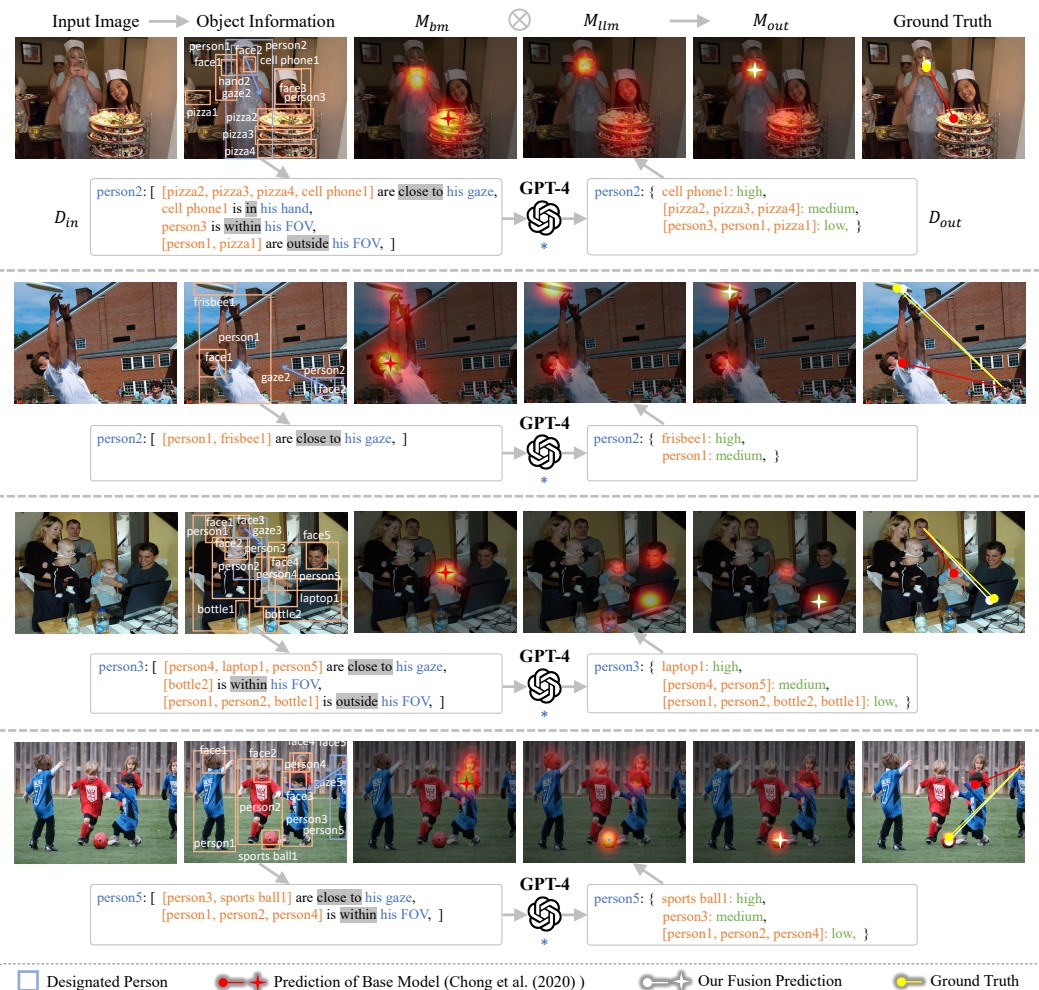

Figure 4: Visualized results of the integrated GTD-LLM framework.

**Long-Tailed Category.** We also evaluate the performance of the integrated GTD-LLM framework in gaze targets of long-tailed categories with a frequency of less than $0.5\%$ in the GazeFollow train set. By integrating with our GTD-LLM module, existing gaze target visual models achieve significant performance improvements in these long-tailed gaze targets in the test set. The recall rate improves by $11\%$–$28\%$, and $L_2$ distance error reduces by $2\%$–$21\%$. These results demonstrate that our GTD-Prompt method can effectively reduce the negative impact of imbalanced distribution of gaze target categories in datasets.

**Small-Sized/Long-Distance Gaze Targets.** For small-sized gaze targets, visual models are easily misled by irrelevant objects with strong saliency. For long-distance gaze targets, visual models struggle to capture the context relationships in the image. Thus, existing gaze target models perform relatively poorly in these challenging scenarios. As shown in Table 2, by integrating with our GTD-LLM module, these shortcomings of existing gaze target visual models are significantly improved. Especially, the smaller the gaze target size or the farther the distance, the more significant the performance improvement. These demonstrate that our GTD-LLM module, which reasons the possible gaze targets from a logical level, can effectively avoid the interference of these irrelevant image features.

**Qualitative Experiments.** Fig. 4 shows the visualized results of the integrated GTD-LLM framework. By integrating with our GTD-LLM module, existing gaze target visual models achieve significant improvements in various challenging scenarios, $e.g.$, activity-related, long-tailed, small-sized, or long-distance gaze targets.

Table 3: Ablation of the task flow prompts $T$ in our GTD-Prompt method.

| $T$ | top-1 | top-3 | top-5 |
|---|---|---|---|
| W/o $t_1$ | 0.782 | 0.873 | 0.901 |
| W/o $t_2$ | 0.724 | 0.836 | 0.875 |
| **Ours** | **0.824** | **0.907** | **0.932** |

Table 4: Ablation of the position relationship rules $R$ in our GTD-Prompt method.

| $R$ | top-1 | top-3 | top-5 |
|---|---|---|---|
| W/o $r_a$ | 0.327 | 0.529 | 0.596 |
| W/o $r_d$ | 0.760 | 0.865 | 0.898 |
| **Ours** | **0.824** | **0.907** | **0.932** |

Table 5: Ablation of the peak value $A$ corresponding to the predicted target probabilities.

| $A$ | | | Video + GTD-LLM Avg. Dist. ↓ | Yang* + GTD-LLM Avg. Dist. ↓ |
|---|---|---|---|---|
| 1.0 | 0.7 | 0.4 | 0.131 | 0.118 |
| 1.0 | 0.1 | 0.0 | 0.130 | 0.117 |
| **1.0** | **0.3** | **0.1** | **0.128** | **0.115** |

Table 6: Ablation of the bias $b_{llm}$ in the multi-hot target heatmap $M_{llm}$.

| $b_{llm}$ | Video + GTD-LLM Avg. Dist. ↓ | Yang* + GTD-LLM Avg. Dist. ↓ |
|---|---|---|
| 0.2 | 0.130 | 0.117 |
| 0.05 | 0.131 | 0.118 |
| **0.1** | **0.128** | **0.115** |

Table 7: Ablation of the threshold $\beta$ in the distance relationship rule $r_d$.

| $\beta$ | top-1 | top-3 | top-5 |
|---|---|---|---|
| $0.25w_h$ | 0.786 | 0.882 | 0.913 |
| $0.75w_h$ | 0.798 | 0.890 | 0.921 |
| $w_h$ | 0.781 | 0.875 | 0.906 |
| **$0.5w_h$** | **0.824** | **0.907** | **0.932** |

Table 8: Ablation of the thresholds $\alpha_1$ and $\alpha_2$ in the angular relationship rule $r_a$.

| $\alpha_1$ | $\alpha_2$ | top-1 | top-3 | top-5 |
|---|---|---|---|---|
| 10° | 45° | 0.802 | 0.893 | 0.921 |
| 20° | 45° | 0.807 | 0.896 | 0.923 |
| 15° | 30° | 0.786 | 0.883 | 0.914 |
| 15° | 60° | 0.795 | 0.889 | 0.918 |
| **15°** | **45°** | **0.824** | **0.907** | **0.932** |

## 4.3 DOMAIN ADAPTATION

As shown in Table 2, we provide the experimental results of the integrated GTD-LLM framework in the complete gaze target datasets. By integrating with our GTD-LLM module, all these gaze target visual models achieve universal performance improvements in the complete GazeFollow test set and VideoAttnTarget test set. We also evaluate the domain adaptation performance of the integrated GTD-LLM framework across different gaze target datasets. Considering that the GazeFollow dataset contains richer scenes and gaze targets, we use it as the source domain $D_1$. Then, the VideoAttnTarget dataset is set as the target domain $D_2$. $D_1 \rightarrow D_2$ represents integrating our GTD-LLM module with existing gaze target visual models which are only trained in the source domain, and let them reason in the target domain directly. By integrating with our GTD-LLM module, all these gaze target visual models achieve significant performance improvements in the target domain with the $L_2$ distance error reducing by $4\%$–$13\%$. These results demonstrate that our method can effectively improve the domain adaptation ability of existing visual models across different gaze target datasets.

## 4.4 ABLATION STUDY

We conduct a series of ablation experiments in the GazeFollow test set to validate the effectiveness of the integrated GTD-LLM framework. Due to the guidance of GPT-4 for multi-hot prediction of gaze targets in our GTD-Prompt method, we use the common used Top-N Accuracy metric to evaluate the prediction accuracy and fault tolerance of our GTD-LLM module. This metric indicates whether the Top-N most likely gaze targets predicted by GPT contain the ground truth.

**Ablation of Task Flow Prompts.** As shown in Table 3, we implement several variants of the task flow prompts $T$ in our GTD-Prompt method. 'W/o $t_1$' represents abandoning the instruction *"What kind of scene is this image?"*. 'W/o $t_2$' denotes abandoning the instruction *"For each person, what are they doing?"*. These results demonstrate that the proposed task flow prompts, which decompose gaze target detection into atomic-level tasks, are easier for GPT-4 to understand and reason.

**Ablation of Position Relationship Rules.** As shown in Table 4, we implement several variants of the position relationship rules $R$ in our GTD-Prompt method. 'W/o $r_a$' represents abandoning the angular relationship rule $r_a$, which means only using the distance relationship descriptions. These results demonstrate that without the angular relationship descriptions between objects and human gaze, GPT-4 is difficult to predict the correct gaze target. 'W/o $r_d$' denotes abandoning the distance relationship rule $r_d$, which means only using the angular relationship descriptions. These results demonstrate that the distance relationship descriptions between objects and human hands/feet can help GPT analyze what activities the human is doing, thereby improving the accuracy of gaze target prediction. As shown in Table 8 and 7, we also implement several variants of thresholds $\alpha_1$ and $\alpha_2$ in the angular relationship rule $r_a$, and the threshold $\beta$ in the distance relationship rule $r_d$.

**Ablation of Cross-Modal Fusion Mechanism.** As shown in Table 5, we implement several variants of the peak value $A$ corresponding to the predicted target probabilities $p \in \{$"$high$", "$medium$", "$low$"$\}$. As shown in Table 6, we also conduct ablation experiments on the bias $b_{llm}$ of the multi-hot target heatmap $M_{llm}$ generated by our GTD-LLM module.

Table 9: Evaluation on shared attention detection.

| Method | Accuracy ↑ | $L_2$ Dist. ↓ |
|---|---|---|
| Video Chong et al. (2020) | 83.3 | 57 |
| **Video + GTD-LLM** | **86.5** | **52** |
| Improvement Ratio | 4% | 9% |
| HGTTR Tu et al. (2022) | 90.4 | 46 |
| **HGTTR + GTD-LLM** | **92.7** | **43** |
| Improvement Ratio | 3% | 7% |

Table 10: Evaluation on mutual gaze detection.

| Methods | UCO-LAEO AP ↑ | AVA-LAEO AP ↑ |
|---|---|---|
| LAEO-Net Marin-Jimenez et al. (2019) | 79.5 | 50.6 |
| **LAEO-Net + GTD-LLM** | **83.0** | **60.4** |
| Improvement Ratio | 4% | 19% |
| MGTR Guo et al. (2022) | 64.8 | 66.2 |
| **MGTR + GTD-LLM** | **68.3** | **69.5** |
| Improvement Ratio | 5% | 5% |

## 4.5 EXPANSION TO DOWNSTREAM TASKS

We conduct a series of experiments to demonstrate that our method can be easily extended to downstream tasks, *e.g.*, shared attention detection and mutual gaze detection, by simply adjusting the corresponding task flow prompts.

**Shared Attention Detection.** This task aims to detect the shared gaze target of multiple people in the image Fan et al. (2018). By adjusting the original task flow prompts $T$, we guide GPT-4 to first perform gaze target detection, and then perform this downstream task and structure its outputs. The detailed explanation is provided in the appendix. As shown in Table 9, by integrating with our GTD-LLM module, these shared attention models achieve universal improvements on the VideoCoAtt benchmark Fan et al. (2018).

**Mutual Gaze Detection.** This is a classification task, aiming to distinguish whether the two designated people in the image are looking at each other Marin-Jimenez et al. (2019). While adjusting the corresponding task flow prompts, we also need to adjust the cross-modal fusion mechanism in the integrated GTD-LLM framework. The detailed explanation is provided in the appendix. As shown in Table 10, by integrating with our GTD-LLM module, these mutual gaze models achieve universal improvements on the UCO-LAEO and AVA-LAEO benchmarks Marin-Jimenez et al. (2019).

## 5 LIMITATIONS AND BROADER IMPACT

Although GTD-LLM achieves notable improvements in gaze target detection, it still has limitations. The reliance on pre-trained LLMs like GPT-4 introduces a computational overhead during the reasoning phase, which may limit its deployment in real-time applications. In experiments, our GTD-LLM module, which uses GPT-4 as the LLM, takes an average of 0.2 to 2 seconds to complete reasoning on a position relationship dictionary corresponding to an input image. The reasoning speed is affected by the content complexity of the dictionary. The more humans and objects contained in the input image, the longer the LLM reasoning process takes. Meanwhile, this speed is also affected by the latency of GPT's API. This is the current limitation encountered in the engineering of LLMs, and is expected to be solved in the future with the progress of LLM itself. Therefore, these current limitations do not affect our exploration of leveraging LLMs to address gaze target detection in visual scenes. Future work could focus on further enhancing the generalization of the prompt-guided reasoning module across diverse visual tasks. Exploring hybrid approaches that integrate both visual and textual knowledge at a deeper level could further improve gaze target detection performance.

## 6 CONCLUSION

In this paper, we introduced GTD-LLM, the first plug-and-play LLM reasoning module for gaze target detection in visual scenes, providing a new paradigm for traditional pure-visual approaches. The plug-and-play nature of our GTD-LLM module makes it adaptable to any existing gaze target visual models. The integrated GTD-LLM framework effectively bridges the gap between visual data and logical reasoning, universally improving the performance of existing visual models. Through the specially designed prompt engineering method GTD-Prompt, LLMs fully mining the prior knowledge about correlations between human visual attention and daily activities, achieving significant improvements in challenging scenarios. Moreover, its adaptability to downstream tasks, *e.g.*, shared attention detection and mutual gaze detection, further underscores the versatility of the proposed method. Our work offers a new avenue for integrating LLMs into visual reasoning tasks. Future work will extend our approach to other complex visual reasoning tasks.

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

# A APPENDIX

**Why not feed the object coordinates into GPT?** If we directly feed these coordinates as input to GPT-4, it will bring a huge reasoning burden to GPT-4 in understanding object position relationships. When there are many object and human instances in the image, this burden will significantly increase. Besides, this may also lead to some misunderstandings of object position relationships in GPT-4, especially when analyzing angular relationships.

**Why not feed the calculated angle/distance values into GPT?** Although this can avoid the above problems, we find that in many cases GPT-4 still struggles to understand the logical relationships between these angle/distance values and human visual attention. Therefore, we consider how to transform these angle/distance values into natural language descriptions which are easier for GPT-4 to understand.

**Why need to structure the position relationship descriptions?** Through the above operation, we create $2 \times M \times (N - 1)$ position relationship descriptions for each image, where $M$ denotes the number of human instances, $N$ denotes the number of all object and human instances. When $M$ and $N$ are relatively large, directly inputting these natural language descriptions into GPT-4 will cause the input context to be too long, increasing the reasoning burden of GPT-4.

**Why need to decompose the gaze target detection task?** Directly having GPT-4 reason each human's gaze targets, may still lead to GPT-4 ignoring the correlations between human visual attention and daily activities, resulting in incorrect predictions. Therefore, we consider how to decompose the gaze target detection task into atomic-level tasks, which are more easier for GPT-4 to understand and reason.

**Why not let GPT-4 reason the out-of-frame classification task?** Due to the excellent performance of existing gaze target models in this classification task, up to $0.944$ on the AP metric Tonini et al. (2023), we do not make GPT-4 reason whether the gaze target is located within or outside the image. Besides, according to common sense, human visual attention may be focused on the objects which are difficult to detect, *e.g.*, walls, sky, ground, *etc.*. Therefore, using GPT-4 reason the out-of-frame classification task from the detected objects may result in errors.

Task Flow Prompts $T_{sa} = \{t_1, t_2, t_3, t_{sa}, t'_{sa}\}$

$t_1$: What kind of scene is this image?

$t_2$: For each person, what are they doing?

$t_3$: For each person, where are they looking?

$t_{sa}$: Is there multiple people looking at the same target?

$t'_{sa}$: If there is, please output the shared gaze target and the corresponding people in the following format:
{
    [obj1, person1, person2, ...]: high,
    [obj2, person3, person4, ...]: medium,
    ...
}

**Instruction** (Role: System)

Figure 5: Adjusted task flow prompts $T_{sa}$ for shared attention detection.

Task Flow Prompts $T_{mg} = \{t_1, t_2, t_3, t_{mg}, t'_{mg}\}$

$t_1$: What kind of scene is this image?

$t_2$: For each person, what are they doing?

$t_3$: For each person, where are they looking?

$t_{mg}$: Is there anyone looking at each other?

$t'_{mg}$: If there is, please output the people who look at each other and the corresponding probabilities in the following format:
{
    [person1, person2]: high,
    [person3, person4]: medium,
    ...
    ...
}

**Instruction** (Role: System)

Figure 6: Adjusted task flow prompts $T_{mg}$ for mutual gaze detection.

**Why abandon the AUC metric?** Due to the pixel multiplication operation performed on the multi-hot target heatmap output by our GTD-LLM module and the single-hot heatmap output by existing gaze target visual models, the final fusion heatmap no longer follows a two-dimensional Gaussian distribution like the ground-truth target heatmap which is generated from the annotated gaze points. Therefore, calculating the similarity between them, *i.e.*, the area under curve (AUC) metric, is not appropriate. Besides, we also abandon the AP metric for out-of-frame classification in the VideoAttnTarget benchmark, since we do not have GPT-4 reason this task.

**Adjustment of the Task Flow Prompts in Shared Attention Detection.** The adjusted task flow prompts $T_{sa}$ in this task is shown in Fig. 5. We use $T_{sa}$ to guide GPT-4 to reason the shared gaze target and the corresponding people. Then, we transform the predictions of GPT-4 into heatmaps through our modal transformation mechanism, and integrate them with the output of existing shared attention visual models through our fusion mechanism in a plug-and-play manner.

**Adjustment of the Task Flow Prompts in Mutual Gaze Detection.** The adjusted task flow prompts $T_{mg}$ in this task is shown in Fig. 6. We use $T_{mg}$ to guide GPT-4 to reason the people who are looking at each other. Then, we transform the qualitative predictions of GPT-4 into quantitative confidence scores, and integrate them with the confidence score output by existing mutual gaze visual models through a simple multiplication operation to obtain the final classification prediction.

