# OpenReview forum: "GTD-LLM: A Plug-and-Play LLM Reasoning Module for Gaze Target Detection"
_ICLR.cc/2025/Conference — ICLR 2025 Conference Withdrawn Submission_

### Official Review · Reviewer_7K3N · 2024-10-23

**Soundness:** 2
**Presentation:** 2
**Contribution:** 2
**Rating:** 3
**Confidence:** 4

**Summary:**

Inspired by the success of large language models, the authors propose to incorporate both conventional visual-based methods with language-based methods to improve the gaze target detection task. The proposed GTD-LLM module is of plug-and-play spirit and can be combined with existing gaze target detection methods. Specifically, the GTD-LLM module formulates the gaze target detection method as a target gaze object detection and aims to localize the key objects among all detected objects. To achieve that, the GTD-LLM module consists of prompt engineering method GTD-Prompt, to guide LLMs like GPT-4 to perform logical reasoning on possible gaze objects, without the need for any training or fine-tuning. Instead, the predictions from existing gaze target detection methods are fed to GTD-LLM to guide its prediction. Then the target object, or the key object, is converted to the probability map and combined with existing gaze prediction methods to obtain more accurate results.

The authors prepare a new sub-set from existing gaze target detection datasets, and focus on mainly the object-level predictions. Results on the subset demonstrate the effectiveness of introducing the plug-and-play GPT-prompt based module in gaze prediction.

**Strengths:**

1. The plug-and-play method is user-friendly and enhances overall performance when used alongside state-of-the-art gaze target detection techniques.
2. The proposed method is straight-forward and easy to comprehend.
3. The concept of integrating GPT to enhance the reasoning process is intriguing.

**Weaknesses:**

1. The proposed method is heavily based on existing techniques, resulting in an incremental approach overall.
2. The abstract's claim that existing methods “detect salient objects” is exaggerated (l.015); for instance, Bao's method (l.039) does not mention any “object” information.
3. The assumption that gaze targets consistently align with objects is overly strong and appears to be tailored to specific datasets, making the object-level detection module less versatile and narrowing the predictive capability of the proposed method.
4. There are numerous hyperparameters and heuristics involved; for example, setting beta in Eq. 2 to “0.5 times the width of the human face” to assess the proximity of other objects to the hand seems poorly justified and unreasonable.
5. The inclusion of a position relationship dictionary seems unnecessary; since the authors utilize GPT-4, this information could be captured in a VQA-like format. They could query GPT-4 about relationships and restrict responses to “in,” “near,” or “far from” when describing object relationships.
6. The overall prediction process lacks interpretability, particularly when converting language descriptions, such as “p ∈ {‘high,’ ‘medium,’ ‘low’}” (l.251), into probabilities.

**Questions:**

Please provide your answers to the following extended questions:

1. The proposed method heavily relies on existing techniques, resulting in an incremental approach. There are more systematic ways to enhance the reasoning process in human gaze target estimation, such as framing the sequence of queries as a Chain-of-Thought. What makes this design superior? Merely combining existing methods leans more toward engineering than genuine research.
2. The claim in the abstract that existing methods “detect salient objects” is exaggerated (l.015); for example, Bao's method (l.039) does not reference any “object” information. The authors should clarify this point.
3. The assumption that gaze targets consistently correspond to objects is overly simplistic and appears tailored to specific datasets, reducing the versatility of the object-level detection module and limiting the proposed method’s predictive capability. Given that the authors design experiments based on the concept of "object," they should justify why introducing this concept is necessary without imposing additional constraints on the task.
4. The presence of several hyperparameters and heuristics raises concerns; for instance, setting beta in Eq. 2 to “0.5 times the width of the human face” to evaluate the proximity of other objects to the hand seems inadequately justified. The authors should provide more details and justifications for these design choices.
5. The necessity of including a position relationship dictionary is questionable; since the authors employ GPT-4, this information could be effectively represented in a VQA-like format. They could query other vision-language models about relationships and limit responses to “in,” “near,” or “far from” when describing object relationships. The authors should better justify their design decisions.
6. The overall prediction process lacks interpretability, particularly in converting language descriptions like “p ∈ {‘high,’ ‘medium,’ ‘low’}” (l.251) into probabilities. What is the rationale behind this design?
7. Please follow the conventional gaze target detection methods and report the generic results (AUC, Dist, Ang) on the full test set of Gazefollow. The title of this paper is "Gaze Target Detection", not "Target Gaze Object Detection".

---

### Official Review · Reviewer_9bjt · 2024-10-30

**Soundness:** 2
**Presentation:** 2
**Contribution:** 3
**Rating:** 5
**Confidence:** 4

**Summary:**

This paper studies the task of gaze target detection (GTD), which aims to predict where people in an image are looking. Given that the generalizability of previous work is limited by relying solely on visual information, this paper introduces a plug-and-play LLM-based reasoning module to enhance logical reasoning on gaze targets for GTD. Extensive experiments are conducted on public datasets and methods to validate the model's effectiveness.

**Strengths:**

1. This paper explores the a plug-and-play LLM module for gaze target detection to close the gap between visual data and logical resoning. This is a novel inverstigation in gaze target detection.
2. This paper transforms the relationship between objects and people into a description, using LLM for inference.
3. This paper evaluates the proposed GTD-LLM on multiple methods.

**Weaknesses:**

1. The configuration of COCO-category gaze targets is unclear, with no data statistics or introduction. For example, what is the distrbution of COCO-LT?
2. The parameters of $\\alpha_1$, $\\alpha_2$ and $\\beta$ appear to heuristic and sensitive, as shown in the ablation study in Tables 7 and 8.
3. The paper only reports the experimental results on shared attention detection and mutual gaze detection, faling to provide the visulization results.
4. This paper failed to provided the failure cases to analysis the limitations.

**Questions:**

1. What is the impact of gaze direction on the accuracy of GTD? If the model adopts the gaze direction predicted by the baseline model, will the performance decrease? This paper adopts a pre-trained model to estimate the gaze direction, which seems to introduce a strong prior for GTD. In other words, this reduces the challenge of detecting the gaze target. In previous work, modules are usually designed to estimate the gaze direction.
2. In the first example of Figure 4, the predicted gaze target is **cell phone1**, while the groun truth is **pizza 2**. This result reflects that the proposed GTD-Prompt fails to reason about the environment when there are multiple potential targets.
3. From the 2-4 exampels in Figure 4, we can see it that the prompt **[xx, xx] are close to his gaze** contains only **one object** and **multiple person**. Interstingly, **one object** is both the predicted gaze target and the ground truth. Does the position relationship based prompt design proposed in this paper reduce the difficulty of inference in GTD, making it easier for the model to predict the results?

---

### Official Review · Reviewer_3JJx · 2024-11-01

**Soundness:** 3
**Presentation:** 2
**Contribution:** 2
**Rating:** 5
**Confidence:** 3

**Summary:**

This paper proposed a plug-and-play GTD_LLM approach for the traditional visual gaze detection task. The starting point lies in the fact that a great deal of prior knowledge is required to accurately identify gaze. Consequently, they made use of LLMs. The authors proposed three steps (object information extraction, object position analysis, and gaze target reasoning) to construct the prompts for simulating the human thought process. This GTD-LLM can adapt gaze target detection tasks and adjust them to be referenced in other downstream tasks.

**Strengths:**

1.	The LLM-based approach was first tried on gaze target detection and is novel.
2.	The design of human thought processes seems reasonable for the gaze target detection.
3.	Experimental and visualization results validate the effectiveness of the GTD_LLM plug.

**Weaknesses:**

1.	GPT-4's input has no visual content, only processed textual data. But I think it would be better in the gaze target detection if take visual content, right?
2.	The ablation of LLMs is missing. I wonder how much the performance boost this LLMs contributes by reasoning the input as p ∈ {" high ", "medium", "low"}.
3.	The specific instruction does not appear in the paper. The contribution of article is prompts, and I feel that publish specific prompts will better attract the reader's attention.

**Questions:**

1.	The instruction to GPT-4 seems only textual data, without visual information. Can Authors explain why this improves the prior knowledge of mining current visual?
2.	If do not use LLM reasoning, will there be a huge drop in method performance with only the Position Relationship Rules?
3.	Will the specific the instruction (‘Role: System’) be published, in the section of “Task Flow Prompts”?
4.	In the future work, if objects outside the scene, it may also be a good direction. Can GTD_LLM solve the problem?

---

### Official Review · Reviewer_YLcW · 2024-11-04

**Soundness:** 1
**Presentation:** 2
**Contribution:** 1
**Rating:** 1
**Confidence:** 4

**Summary:**

This paper proposes a new framework for gaze target detection, consisting of multiple off-the-shelf computer vision models and a new LLM module that connects these vision models. The core contribution of the paper is the prompt engineering design of the LLM module. The authors show that plugging in their LLM module, along with existing gaze target detection models, into their framework, improves the target detection performance over the baselines compared in the paper.

**Strengths:**

It is the first paper that leverages LLM-based reasoning for gaze target detection, and the result shows that the proposed prompt engineering method, in conjunction with other off-the-shelf computer vision models, can achieve considerable improvement in gaze target detection performance.

It shows that the method can also be adapted to other related downstream tasks such as mutual gaze and shared attention tasks.

**Weaknesses:**

The overall system overly engineered and contrived. I understand that they wanted to design a plug-an-play module which can be applied on top of the existing models. However, this new module is not a just simple operation, a new loss function, or a new training regime. The newly added module is a whole GPT-4 with a bit of prompt engineering. Undoubtedly, the end results will perform much better than all the baselines that operates without LLMs. Moreover, the proposed system uses many other off-the-shelf CV models (MM-GroundingDINO for object detection, OpenPose for pose detection, L2cs-net for gaze direction) in addition to the LLM and gaze target detection model. All of these additional variables make it hard to assess the true value of their method; How does one distinguish whether the improved results are simply coming from the power of LLM itself, and/or from the other object/pose/gaze detector that are also in use? In order to demonstrate the value of the proposed design fairly, they need to conduct much more systematic and extensive evaluations.

By nature, LLMs are more general and powerful than the traditional pure-visual approaches. If so, why do we still such a complicated pipeline (object/pose/gaze detector + LLM + gaze target model)? We could possibly achieve similar output only with LLMs (or VLMs) without the need for pure vision models using a bit different prompt engineering. Have the authors tried alternative prompts with LLMs that do not require running existing gaze target models and object/pose/gaze detectors? The current design seems rather for the sake of making the method to be usable on top of the existing gaze target models which may not be needed at all in practice.

Moreover, even if all of the claims of the paper were true, the fact that this requires running LLM at inference time makes it less appealing for practical use. The cost of inference should be taken into account as well when comparing with the pure-visual models that are being benchmarked in the paper.

Additionally, I see limitations of the method as a gaze target detector on its own.

The paper proposes to do gaze target detection at object-level rather than coordinate-level as in previous works. The underlying motivation for this is understandable but this is artificially limiting the granularity and category of the output. How can we tell when a person is looking at certain part of the object vs the opposite part of the same object?

In 643-645 "Due to the excellent performance of existing gaze target models in this classification task, up to 0.944 on the AP metric, we do not make GPT-4 reason whether the gaze target is located within or outside the image. Besides, according to common sense, human visual attention may be focused on the objects ..". I do not find the answers convincing at all for why they did not share the performance of the classification task. The fact that the performance on this metric is saturated should not be the reason for not reporting the performance on it. At the very least, they need to show that their method can do just as well as the existing methods.

Based on these limitations, the method in this paper is actually more limited (not generalizable) than existing method in gaze target detection task.

**Questions:**

Most of the questions are included in the above. A few additional ones are

Have the authors have considered using VLMs instead of LLMs? If so, why or why not?

Figure 2. needs to be clarified. What does * and solid/transparent color mean? What does the x operation mean? These should be illustrated as a legend or described as a text in caption.

---

### Official Review · Reviewer_aCKo · 2024-11-06

**Soundness:** 2
**Presentation:** 3
**Contribution:** 2
**Rating:** 3
**Confidence:** 5

**Summary:**

This paper attempts to augment the performance of existing gaze target detection methods by incorporating information obtained from LLMs. This is achieved by separately detecting potential gaze targets using off-the-shelf detectors and then utilizing prompt engineering to perform reasoning about possible gaze targets using off-the-shelf LLMs. The LLM model produces an independent prediction about possible gaze locations which is then fused with existing gaze prediction models using a standard pixelwise multiplication. The experiments demonstrate that LLMs contain information which is relevant for gaze target prediction, since the fused prediction accuracy improves upon the baseline models by varying percentages. However, the experimental comparison is not straightforward since the LLM approach cannot support the standard metrics for which the existing gaze models are trained, making a direct assessment of improvement more difficult. The method is tested on a subset of existing gaze prediction models for which source code is available.

**Strengths:**

The strengths of the paper are:
- An novel prompt engineering approach to leverage existing LLM models for gaze target prediction
- A demonstration that LLMs contain relevant information about gaze target prediction which can be utilized to enhance performance
- Experiments on standard datasets using several existing gaze target prediction models to demonstrate the generalization of the performance gains.

**Weaknesses:**

There are multiple significant weaknesses. The first is the fact that the LLM approach does not support the standard gaze target prediction problem. In particular, the LLM does not provide an in/out prediction, which indicates whether or not the gaze target is visible in the frame. This is a crucial part of the prediction task, and it is unclear why LLMs cannot be leveraged effectively given that they can be used for gaze reasoning. Furthermore, the use of object detectors and the creation of object-based predictions changes the problem formulation. It is unclear how this impacts complex gaze targets. For objects with significant spatial extent, such as humans and animals, it may not be possible to generate detection hypotheses at a sufficiently-fine grained spatial resolution, and this in turn could be a source of error. A strength of existing saliency-based formulations is that performance does not depend on the generalization performance of object detectors. A related issue is the inability to use LLMs to produce improvements in AUC, which is the metric which most effectively captures gaze prediction performance, particularly in complex scenes. A second set of limitations has to do with the experimental evaluations. Most of the performance gains arise for older models which are no longer SOTA. The following competitive methods have code available but are not included in the experiments. It is unclear how well the LLM approach performs on its own, which would seem to be a useful baseline. It is not really surprising that LLMs contain useful information about gaze targets given their applicability to a broad range of tasks. It would be valuable to understand what kinds of cues LLM can exploit. For example, modern architectures incorporate modules for depth, eye gaze estimation, body pose, etc. which are shown to produce gains. To what extent are LLMs also leveraging these cues? The use of LLMs is somewhat limited and does not provide a lot of insight into what they can do in detail for this task.

Gupta, A., Tafasca, S., Odobez, J.M.: A modular multimodal architecture for gaze target prediction: Application to privacy-sensitive settings. In: Proceedings of the IEEE/CVF Conference on Computer Vision and Pattern Recognition Workshops. pp. 5041–5050 (2022)

Miao, Q., Hoai, M., Samaras, D.: Patch-level gaze distribution prediction for gaze following. In: Proceedings of the IEEE/CVF Winter Conference on Applications of Computer Vision. pp. 880–889 (2023)

**Questions:**

- What is performance of LLM alone as a baseline?
- How is the heatmap generated from the LLM output in detail (e.g. what determines how many modes it will have)?
- Given that the LLM can't make in/out determination how is accuracy computed in the experiments exactly?
- How were metrics like RR obtained for the existing models prior to the fusion with LLM?
- What are the performance gains for the methods Gupta et al and Miao et al.?

---

### Note · Authors · 2024-11-12

I have read and agree with the venue's withdrawal policy on behalf of myself and my co-authors.